# TARGETED VAE: STRUCTURED INFERENCE AND TARGETED LEARNING FOR CAUSAL PARAMETER ESTIMATION

## ABSTRACT

Undertaking causal inference with observational data is extremely useful across a wide range of domains including the development of medical treatments, advertisements and marketing, and policy making. There are two main challenges associated with undertaking causal inference using observational data: treatment assignment heterogeneity (i.e., differences between the treated and untreated groups), and an absence of counterfactual data (i.e. not knowing what would have happened if an individual who did get treatment, were instead to have not been treated). We address these two challenges by combining structured inference and targeted learning. To our knowledge, Targeted Variational AutoEncoder (TVAE) is the first method to incorporate targeted learning into deep latent variable models. Results demonstrate competitive and state of the art performance.

## 1 INTRODUCTION

The estimation of the causal effects of interventions or treatments on outcomes is of the upmost importance across a range of decision making processes and scientific endeavours, such as policy making (Kreif & DiazOrdaz, 2019), advertisement (Bottou et al., 2013), the development of medical treatments (Petersen et al., 2017), the evaluation of evidence within legal frameworks (Pearl, 2009; Siegerink et al., 2016) and social science (Vowels, 2020; Hernan, 2018; Grosz et al., 2020). Despite the common preference for Randomized Controlled Trial (RCT) data over observational data, this preference is not always justified. Besides the lower cost and fewer ethical concerns, observational data may provide a number of statistical advantages including greater statistical power and increased generalizability (Deaton & Cartwright, 2018). However, there are two main challenges when dealing with observational data. Firstly, the group that receives treatment is usually not equivalent to the group that does not (treatment assignment heterogeneity), resulting in selection bias and confounding due to associated covariates. For example, young people may prefer surgery, older people may prefer medication. Secondly, we are unable to directly estimate the causal effect of treatment, because only the factual outcome for a given treatment assignment is available. In other words, we do not have the counterfactual associated with the outcome for a different treatment assignment to that which was given. Treatment effect inference with observational data is concerned with finding ways to estimate the causal effect by considering the expected differences between factual and counterfactual outcomes.

We seek to address the two challenges by proposing a method that incorporates targeted learning techniques into a disentangled variational latent model, trained according to the approximate maximum likelihood paradigm. Doing so enables us to estimate the expected treatment effects, as well as individual-level treatment effects. Estimating the latter is especially important for treatments that interact with patient attributes, whilst also being crucial for enabling individualized treatment assignment. Thus, we propose the Targeted Variational AutoEncoder (TVAE), undertake an ablation study, and compare our method's performance against current alternatives on two benchmark datasets.

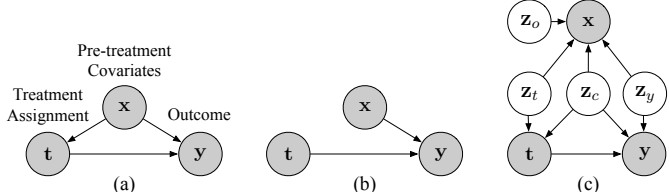

Figure 1: Directed Acyclic Graphs (DAGs) for (a) the problem of estimating the effect of treatment $\mathbf{t}$ on outcome $\mathbf{y}$ with confounders $\mathbf{x}$. DAG (b) reflects an RCT. DAG (c) illustrates TVAE and is an extension of the DAG by Zhang et al. (2020), where the structure is *a priori* assumed to factorize into into risk $\mathbf{z}_y$, instrumental $\mathbf{z}_t$, and confounding factors $\mathbf{z}_c$. We extend their model with $\mathbf{z}_o$ to account for the scenario whereby not all covariates will be related to treatment and/or outcome.

## 2 BACKGROUND

**Problem Formulation:** A characterization of the problem of causal inference with no unobserved confounders is depicted in the Directed Acyclic Graphs (DAGs) shown in Figs. 1(a) and 1(b). Fig. 1(a) is characteristic of observational data, where the assignment of treatment is related to the covariates. Fig. 1(b) is characteristic of the ideal RCT, where the treatment is unrelated to the covariates. Here, $\mathbf{x}_i \sim p(\mathbf{x}) \in \mathbb{R}^m$ represents the $m$-dimensional, pre-treatment covariates for individual $i$ assigned factual treatment $t_i \sim p(\mathbf{t}|\mathbf{x})$ resulting in factual outcome $y_i^t \sim p(\mathbf{y}|\mathbf{x}, \mathbf{t})$. Together, these constitute dataset $\mathcal{D} = \{[y_i, t_i, \mathbf{x}_i]\}_{i=1}^N$ where $N$ is the sample size.

The conditional average treatment effect for an individual with covariates $\mathbf{x}_i$ may be estimated as $\hat{\tau}_i(\mathbf{x}_i) = \mathbb{E}[y_i|\mathbf{x}_i, do(\mathbf{t} = 1) - y_i|\mathbf{x}_i, do(\mathbf{t} = 0)]$, where the expectation accounts for the non-determinism of the outcome (Jesson et al., 2020). Alternatively, by comparing the post-intervention distributions when we intervene on treatment $\mathbf{t}$, the Average Treatment Effect (ATE) is $\hat{\tau}(\mathbf{x}) = \mathbb{E}_\mathbf{x}[\mathbb{E}[\mathbf{y}|\mathbf{x}, do(\mathbf{t} = 1)] - \mathbb{E}[\mathbf{y}|\mathbf{x}, do(\mathbf{t} = 0)]]$. Here, $do(\mathbf{t})$ indicates the intervention on $\mathbf{t}$, setting all instances to a static value, dynamic value, or distribution and therefore removing any dependencies it originally had (Pearl, 2009; van der Laan & Rose, 2018; 2011). This scenario corresponds with the DAG in Fig. 1(b), where treatment $\mathbf{t}$ is no longer a function of the covariates $\mathbf{x}$.

Using an estimator for the conditional mean $Q(\mathbf{t}, \mathbf{x}) = \mathbb{E}(\mathbf{y}|\mathbf{t}, \mathbf{x})$, we can calculate the Average Treatment Effect (ATE) and the empirical error for estimation of the ATE (eATE).[1] In order to estimate eATE we assume access to the ground truth treatment effect $\tau$, which is only possible with synthetic or semi-synthetic datasets. The Conditional Average Treatment Effect (CATE) may also be calculated and the Precision in Estimating Heterogeneous Effect (PEHE) is one way to evaluate a model's efficacy in estimating this quantity. See the appendix for the complete definitions of these terms.

**The Naive Approach:** The DAG in Fig. 1(a) highlights the problem with taking a naive approach to modeling the joint distribution $p(\mathbf{y}, \mathbf{t}, \mathbf{x})$. The structural relationship $\mathbf{t} \leftarrow \mathbf{x} \rightarrow \mathbf{y}$ indicates both that the assignment of treatment $\mathbf{t}$ is dependent on the covariates $\mathbf{x}$, and that a backdoor path exists through $\mathbf{x}$ to $\mathbf{y}$. In addition to our previous assumptions, if we also assume linearity, adjusting for this backdoor path is a simple matter of adjusting for $\mathbf{x}$ by including it in a logistic regression. The naive method is an example of the uppermost methods depicted in Fig. 2, and leads to the largest bias. The problem with the approach is (a) that the graph is likely misspecified such that the true relationships between covariates as well as the relationships between covariates and the outcome may be more complex. There is also problem (b), that linearity is not sufficient to 'let the data speak' (van der Laan & Rose, 2011) or to avoid biased parameter estimates. Using powerful nonparametric models (e.g., neural networks) may solve the limitations associated with linearity and interactions to yield a consistent estimator for $p(\mathbf{y}|\mathbf{x})$, and such a model is an example of the middlemost methods depicted in Fig. 2. However, this estimator is not targeted to the estimation of the causal effect parameter $\tau$, only predicting the outcome, and we require a means to reduce residual bias.

**Targeted Learning:** Targeted Maximum Likelihood Estimation (TMLE) (Schuler & Rose, 2016; van der Laan & Rose, 2011; 2018; van der Laan & Starmans, 2014) falls under the lowermost

---

[1] For a binary outcome variable $\mathbf{y} \in \{0, 1\}$, $\mathbb{E}(\mathbf{y}|\mathbf{t}, \mathbf{x})$ is the same as the conditional probability distribution $p(\mathbf{y}|\mathbf{t}, \mathbf{x})$.

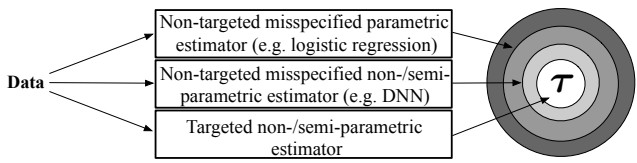

Figure 2: Different methods for estimating the causal parameter $\tau$ yield different levels of bias. Adapted from (van der Laan & Rose, 2011).

methods depicted in Fig. 2 and follows an approach involving three main steps: (1) estimation of the conditional mean $\mathbb{E}(\mathbf{y}|\mathbf{t}, \mathbf{x})$ with estimator $Q^0(\mathbf{t}, \mathbf{x})$, (2) estimation of the propensity scores with estimator $g(\mathbf{t}|\mathbf{x})$, and (3) updating the conditional mean estimator $Q^0$ to get $Q^*$ using the propensity scores to attain an estimate for the causal parameter $\tau$.

The propensity score for individual $i$ is defined as the conditional probability of being assigned treatment $g(t_i, \mathbf{x}_i) = p(\mathbf{t} = t_i|\mathbf{x} = \mathbf{x}_i), \in [0, 1]$ (Rosenbaum & Rubin, 1983). The scores can be used to compensate for the relationship between the covariates and the treatment assignment using Inverse Probability of Treatment Weights (IPTWs), reweighting each sample according to its propensity score. Step (3) is undertaken using 'clever covariates' which are similar to the IPTWs. They form an additional covariate variable $H(1, \mathbf{x}_i) = g(1|\mathbf{x}_i)^{-1}$ for individual $i$ assigned treatment, and $H(0, \mathbf{x}_i) = -g(1|\mathbf{x}_i)^{-1}$ for individual $i$ not assigned treatment. Note that when we condition on a single numeric value we imply an intervention (e.g. $g(1|\mathbf{x}_i) \equiv g(do(\mathbf{t} = 1)|\mathbf{x}_i)$). A logistic regression is then undertaken as $\mathbf{y} = \sigma^{-1}[Q^0(\mathbf{t}, \mathbf{x})] + \epsilon H(\mathbf{t}, \mathbf{x})$ where $\sigma^{-1}$ is the logit/inverse sigmoid function, $Q^0(\mathbf{t}, \mathbf{x})$ is set to be a constant, suppressed offset and $\epsilon$ represents a fluctuation parameter which is to be estimated from the regression. Once $\epsilon$ has been estimated, we acquire an updated estimator:

$$Q^1(do(\mathbf{t} = t), \mathbf{x}) = \sigma\left[\sigma^{-1}[Q^0(\mathbf{t}, \mathbf{x})] + \epsilon H(\mathbf{t}, \mathbf{x})\right] \tag{1}$$

This equation tells us that our new estimator $Q^1$ is equal to the old estimator balanced by the corrective $\epsilon H(\mathbf{t}, \mathbf{x})$ term. This term adjusts for the bias associated with the propensity scores. When the $\epsilon$ parameter is zero, it means that there is no longer any influence from the 'clever covariates' $H()$. The updated estimator $Q^1$ can then be plugged into the estimator for $\hat{\tau}(Q^1; \mathbf{x})$. When the optimal solution is reached (i.e. when $\epsilon = 0$), the estimator also satisfies what is known as the efficient Influence Curve (IC), or canonical gradient equation (Hampel, 1974; van der Laan & Rose, 2011; Kennedy, 2016):

$$\sum_{i=1}^{N} IC^*(y_i, t_i, \mathbf{x}_i) = 0 = \sum_{i=1}^{N}\left[H(t_i, \mathbf{x}_i)(y_i - Q(t_i, \mathbf{x}_i)) + Q(1, \mathbf{x}_i) - Q(0, \mathbf{x}_i) - \tau(Q; \mathbf{x})\right] \tag{2}$$

where $IC(y_i, t_i, \mathbf{x}_i)$ represents the IC, and $IC^*(y_i, t_i, \mathbf{x}_i)$ represents the efficient IC for consistent $Q$ and $g$. It can be seen from the right hand side Eq. 2 that at convergence, the estimator and its estimand are equal: $y_i = Q(t_i, \mathbf{x}_i)$ and $Q(1, \mathbf{x}_i) - Q(0, \mathbf{x}_i) = \tau(Q; \mathbf{x})$. Over the whole dataset, all terms in Eq. 2 'cancel' resulting in the mean $\overline{IC} = 0$. As such, the logistic regression in Eq. 1 represents a solution to the IC via a parametric submodel.

The TMLE method provides a doubly robust, asymptotically efficient estimate of the causal or 'target' parameter, and these theoretical guarantees make it attractive for adaptation into neural networks for causal effect estimation.

## 3 METHODOLOGY

In this section we present the Targeted Variational AutoEncoder (TVAE), a deep generative latent variable model that enables estimation of the average and conditional average treatment effects (ATE and CATE resp.) via the use of amortized variational inference techniques and Targeted Maximum Likelihood Estimation (TMLE). For a review on the relevant VAE theory, see the appendix. A top-

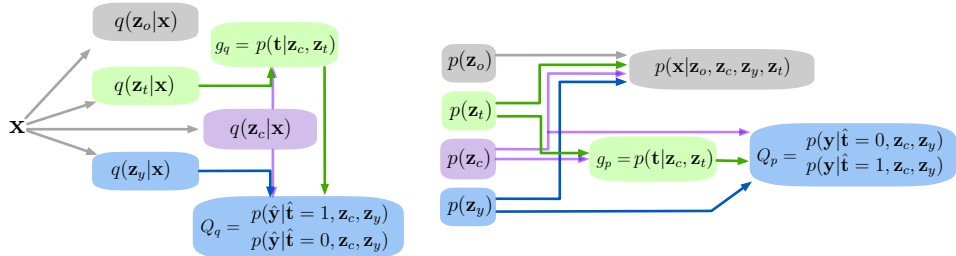

Figure 3: The block-diagram for Targeted VAE. Dashed boxes indicate the variationally inferred latent variables $\mathbf{z}_o$, $\mathbf{z}_c$, $\mathbf{z}_t$, and $\mathbf{z}_y$. Arrows indicate functions, and colors distinguish treatment (green), outcome (blue), covariates (grey), and confounder (purple) related entities.

level diagram for TVAE is shown in Fig. 3 and follows the structure implied by the DAG in Fig. 1(c). A more detailed architectural block-diagram is also presented in the appendix.[2]

***Assumptions:*** As is common (Yao et al., 2020; Guo et al., 2020; Rubin, 2005; Imbens & Rubin, 2015) when undertaking causal inference with observational data, we make three assumptions: (1) Stable Unit Treatment Value Assumption (SUTVA): the potential outcomes for each individual or data unit are independent of the treatments assigned to all other individuals, such that there are no interactions between individuals. (2) Positivity: the assignment of treatment probabilities are all non-zero and non-deterministic $p(\mathbf{t} = t_i | \mathbf{x} = \mathbf{x}_i) > 0$, $\forall$ $\mathbf{t}$ and $\mathbf{x}$. (3) Ignorability: all confounders are observed such that the likelihood of treatment for two individuals with the same covariates is equal, and the potential outcomes for two individuals with the same covariates are also equal $s.t.$ $(\mathbf{y}^{t=1}, \mathbf{y}^{t=0}) \perp\!\!\!\perp \mathbf{t} | \mathbf{x}$ and $\mathbf{t} \perp\!\!\!\perp (\mathbf{y}^{t=1}, \mathbf{y}^{t=0}) | \mathbf{x}$.[3]

***TVAE:*** If one had knowledge of the true causal DAG underlying a set of data, one could undertake causal inference without being concerned for issues relating to structural misspecification. Unfortunately, and this is particularly the case with observational data, we rarely have access to this knowledge. Quite often an observed set of covariates $\mathbf{x}$ are modelled as a group of confounding variables (as per the DAG in Figure 1a). Furthermore, and as noted by Zhang et al. (2020), researchers may be encouraged to incorporate as many covariates into their model as possible, in an attempt to reduce the severity of the ignorability assumption. However, including more covariates than is necessary leads to other problems relating to the curse of dimensionality and (in)efficiency of estimation.

A large set of covariates may be separable into subsets of factors such as instrumental, risk, and confounding factors. Doing so helps us to match our model more closely to the true data generating process, as well as to improve estimation efficiency by 'distilling' our covariate adjustment set. Prior work has explored the potential to discover the relevant confounding covariates via Bayesian networks (Haggstrom, 2017), regularized regression (Kuang et al., 2017), and deep latent variable models based on Variational Autoencoders (VAEs) (Zhang et al., 2020; Louizos et al., 2017). The first two methods *identify* variables (and are variable selection algorithms), whereas VAEs *infer* them, and learn compact, disentangled representations of the observations. The benefit of the latter approach is that it (a) infers latent variables on a datapoint-by-datapoint basis (rather than deriving subsets from population aggregates), (b) under additional assumptions, VAEs have been shown to infer hidden confounders in the presence of noisy proxy variables, thereby potentially reducing the reliance on ignorability) (Louizos et al., 2017), and (c) makes no assumptions about the functional form used to map between covariate and latent space.

We seek to infer and disentangle the latent distribution into subsets of latent factors using VAEs. These latent subsets are $\{\mathbf{z}_t, \mathbf{z}_y, \mathbf{z}_c, \mathbf{z}_o\}$, which represent the instrumental factors on $\mathbf{t}$, the risk factors on $\mathbf{y}$, the confounders on both $\mathbf{t}$ and $\mathbf{y}$, and factors solely related to $\mathbf{x}$, respectively. Without inductive bias, consistently disentangling the latent variables into these factors would be impossible (Locatello et al., 2019). In TVAE this inductive bias is incorporated in a number of ways: firstly, by incorporating supervision and constraining $\mathbf{z}_t$ and $\mathbf{z}_y$ to be predictive of $\mathbf{t}$ and $\mathbf{y}$, respectively; secondly, by constraining $\mathbf{z}_c$ to be predictive of both $\mathbf{t}$ and $\mathbf{y}$; and finally, by employing diagonal-

---

[2]Source code will also made available in supplementary material upon acceptance.
[3]Taken together, assumptions (2) and (3) constitute *strong ignorability*.

covariance priors (isotropic Gaussians) to encourage disentanglement and independence between latent variables. The structural inductive bias on the model is such that $\mathbf{z}_y$, and $\mathbf{z}_t$, and $\mathbf{z}_c$ learn factors relevant to outcome and treatment, for which we provide explicit supervision, thereby leaving $\mathbf{z}_o$ for all remaining factors.

In general, it is impossible to isolate the effect of $\mathbf{t} \to \mathbf{y}$ due to unobserved confounding (D'Amour, 2019), and this is why we make the assumption of ignorability. However, it is worth noting that, under additional assumptions, deep latent variable techniques have been shown to be able to infer hidden confounders from what are known as *noisy proxy* variables present in the dataset (see e.g., Montgomery et al. 2000; Louizos et al. 2017). The assumption of ignorability then shifts from 'all confounders are observed', to 'all unobserved confounders have been inferred from proxies'. Whilst the capability to infer confounders from proxies represents an additional motivation for the use of VAEs, the focus of this work is not to explore whether and by how much we are able to do so, and we therefore maintain the assumption of ignorability.

**Inference:**

$$q(\mathbf{z}_t|\mathbf{x}) = \prod_{d=1}^{D_{z_t}} \mathcal{N}(\mu_d = f_{1d}(\mathbf{x}), \sigma_d^2 = f_{2d}(\mathbf{x})); \ q(\mathbf{z}_y|\mathbf{x}) = \prod_{d=1}^{D_{z_y}} \mathcal{N}(\mu_d = f_{3d}(\mathbf{x}), \sigma_d^2 = f_{4d}(\mathbf{x}))$$

$$q(\mathbf{z}_c|\mathbf{x}) = \prod_{d=1}^{D_{z_c}} \mathcal{N}(\mu_d = f_{5d}(\mathbf{x}), \sigma_d^2 = f_{6d}(\mathbf{x})); q(\mathbf{z}_o|\mathbf{x}) = \prod_{d=1}^{D_{z_o}} \mathcal{N}(\mu_d = f_{7d}(\mathbf{x}), \sigma_d^2 = f_{8d}(\mathbf{x}))$$

$$p(\hat{\mathbf{t}}|\mathbf{z}_t, \mathbf{z}_c) = \text{Bern}(f_9(\mathbf{z}_t, \mathbf{z}_c)) = \text{Bern}(g_q(\mathbf{z}_t, \mathbf{z}_c))$$

$$p(\hat{\mathbf{y}}|\mathbf{z}_y, \mathbf{z}_c, \mathbf{t}) = \text{Bern}(\hat{\mathbf{t}} \cdot f_{10}(\mathbf{z}_y, \mathbf{z}_c) + (1 - \hat{\mathbf{t}}) \cdot f_{11}(\mathbf{z}_y, \mathbf{z}_c)) = \text{Bern}(Q_q(.))$$

**Generation:**

$$p(\mathbf{z}_{\{o,t,c,y\}}) = \prod_d^{D_{\{z_{o,t,c,y}\}}} \mathcal{N}(z_{\{o,t,c,y\}d}|0, 1); \ \ p(\hat{\mathbf{t}}|\mathbf{z}_t, \mathbf{z}_c) = \text{Bern}(h_1(\mathbf{z}_t, \mathbf{z}_c)) = \text{Bern}(g_p(.))$$

$$p(\hat{\mathbf{y}}|\mathbf{z}_y, \mathbf{z}_c, \hat{\mathbf{t}}) = \text{Bern}(\hat{\mathbf{t}} \cdot h_2(\mathbf{z}_y, \mathbf{z}_c) + (1 - \hat{\mathbf{t}}) \cdot h_3(\mathbf{z}_y, \mathbf{z}_c)) = \text{Bern}(Q_p(.))$$

$$p(\hat{\mathbf{x}}_{bin}|\mathbf{z}_c, \mathbf{z}_o, \mathbf{z}_t, \mathbf{z}_y) = \text{Bern}(h_6(\mathbf{z}_c, \mathbf{z}_o, \mathbf{z}_t, \mathbf{z}_y))$$

$$p(\hat{\mathbf{x}}_{cont}|\mathbf{z}_c, \mathbf{z}_o, \mathbf{z}_t, \mathbf{z}_y) = \prod_{d=1}^{D_{x_{cont}}} \mathcal{N}(x_{cont,d}|\mu_d = h_4(\mathbf{z}_c, \mathbf{z}_o, \mathbf{z}_t, \mathbf{z}_y), \sigma_d^2 = h_5(\mathbf{z}_c, \mathbf{z}_o, \mathbf{z}_t, \mathbf{z}_y))$$

$$(3)$$

The proof for identifiability under the assumption of ignorability (or, alternatively, under the assumption that all unobserved confounders have been inferred from proxy variables) has been derived previously by Louizos et al. (2017) and Zhang et al. (2020). The factor $\mathbf{z}_o$ is $d$-separated from $\mathbf{t}$ and $\mathbf{y}$ given $\mathbf{x}$, and does not affect the identification of the causal effect. i.e., $p(y|do(t), \mathbf{x}) = p(y|do(t), \mathbf{z}_{\{t,o,y,c\}}) = p(y|t, \mathbf{z}_y, \mathbf{z}_c)$ (see Zhang et al. 2020 and Louizos et al. 2017). We impose the priors and parameterizations denoted in Equation 3, where $D_{(.)}$ is the number of dimensions in the respective variable (latent or otherwise), and $f_{1-11}$ and $h_{1-6}$ represent fully connected neural network functions. The parameters for these neural networks are learnt via variational Bayesian approximate inference (Kingma & Welling, 2014) according to the following objective:

$$\mathcal{L}_i^{\text{ELBO}} = \sum_i^N \mathbb{E}_{q_c q_t q_y q_o} \left[ \log p\left(\hat{\mathbf{x}}_i|\mathbf{z}_t, \mathbf{z}_c, \mathbf{z}_y, \mathbf{z}_o\right) + \log p\left(\hat{t}_i|\mathbf{z}_t, \mathbf{z}_c\right) + \log p\left(\hat{y}_i|t_i, \mathbf{z}_y, \mathbf{z}_c\right) \right]$$

$$- \left[ D_{KL}\left(q\left(\mathbf{z}_t|\mathbf{x}_i\right) \| p\left(\mathbf{z}_t\right)\right) + D_{KL}\left(q\left(\mathbf{z}_c|\mathbf{x},\right) \| p\left(\mathbf{z}_c\right)\right) \right.$$

$$\left. + D_{KL}\left(q\left(\mathbf{z}_y|\mathbf{x}_i\right) \| p\left(\mathbf{z}_y\right)\right) + D_{KL}\left(q\left(\mathbf{z}_o|\mathbf{x}_i\right) \| p\left(\mathbf{z}_o\right)\right) \right] \tag{4}$$

Note that all Gaussian variance parameterizations are diagonal. In cases where prior knowledge dictates a discrete rather than continuous outcome, equivalent parameterizations to those in Eqs. 3 may be employed. For example, in the IHDP dataset, the outcome data are standardized to have a variance of 1, and the outcome generation model becomes a Gaussian with variance also equal to 1.

Note that separate treatment and outcome classifiers are used both during inference and generation ($Q_q, g_q$ and $Q_p, g_p$ resp.). The classifiers for inference have separate parameters to those use during generation. Predictors or classifiers of outcome incorporate the two-headed approach of (Shalit et al., 2017), and use ground-truth $\mathbf{t}$ are used for $Q_q$ whereas samples $\hat{\mathbf{t}}$ are used for $Q_p$. For unseen test cases, either the ground-truth $\mathbf{t}$ or an sampled treatment $\hat{\mathbf{t}}$ from treatment classifier $g_p$ may be used to simulate an outcome. During training $\hat{\mathbf{t}}$ is used.

We now introduce the targeted regularization, the purpose of which is to encourage the outcome to be independent of the treatment assignment. Following Eq. 1, we define the fluctuation sub-model and corresponding logistic loss for finding $\epsilon$ as:

$$\hat{Q}(g, t_i, \mathbf{z}_i^y, \mathbf{z}_i^c, \epsilon) = \sigma \left[ \sigma^{-1}[Q(t_i, \mathbf{z}_i^y, \mathbf{z}_i^c)] + \epsilon \left( \frac{I(t_i = 1)}{g(t_i = 1; \mathbf{z}_i^t, \mathbf{z}_i^c)} - \frac{I(t_i = 0)}{g(t_i = 0; \mathbf{z}_i^t, \mathbf{z}_i^c)} \right) \right] \quad (5)$$

$$\xi_i(\hat{Q}; \epsilon) = -y_i \log(\hat{Q}(g, t_i, \mathbf{z}_i^y, \mathbf{z}_i^c, \epsilon)) - (1 - y_i) \log(1 - \hat{Q}(g, t_i, \mathbf{z}_i^y, \mathbf{z}_i^c, \epsilon)) \quad (6)$$

In Eq. 5, $I$ is the indicator function. For an unbounded regression loss, mean squared error loss may be used (see appendix). Note that the logistic loss is suitable for continuous outcomes bounded between 0 and 1 (see van der Laan & Rose 2011, pp.121:132 for proof). Putting it all together, we then optimize to find generative parameters for functions $h_{1-6}$, inference parameters for functions $f_{1-12}$, and fluctuation parameter $\epsilon$ as follows:

$$\mathcal{L} = \min \left[ \sum_i^N \left( \mathcal{L}_i^{\text{ELBO}} + \lambda_{TL} \xi_i(Q, g, \epsilon) \right) \right]; \quad \left. \frac{\partial}{\partial \epsilon} \mathcal{L}^* \right|_{\epsilon=0} = \bar{I}\bar{C}^* = 0 \quad (7)$$

Where $\lambda_{TL}$ represents a hyperparameter loss weight for the targeted regularization. At convergence, $\epsilon = 0$ when $Q$ and $g$ become consistent estimators, satisfying the conditions for the EIC (see Eq. 2 and reference van der Laan & Rose 2011, pp125:128).

A further element that differentiates our work from one other recent contribution (Shi et al., 2019) that uses targeted regularization is that the gradients resulting from $\xi$ are *not* taken with respect to $g_p$ or $g_q$ (which are the propensity score arms which we assume to be consistent and unbiased). Targeted learning is concerned with de-biasing the outcome classifier $Q$ using propensity scores from $g$. In other words, assuming the propensity scores are well estimated, the targeted learning regularizer is intended to affect the outcome classifier only, and *not* the propensity score estimator. It is therefore more theoretically aligned (with the targeted learning literature) to apply regularization to the outcome estimator $Q$, and not to $g$. As per Eq. 1, in TMLE, $g$ is assumed to be a consistent estimator, forming part of the de-biasing update process for $Q$, but it is not subject to update itself. In order to prevent the regularization from affecting the propensity arms, the gradients from the regularizer are only taken with respect to all parameters that influence this outcome classifier (which include upstream parameters for $Q_q$ as well as the more direct parameters $Q_p$). We use Pytorch's 'detach' method on the propensity scores when calculating the targeted regularization. This method decouples the propensity score arm from backpropagation relating to the computation of the regularization value. In contrast, Dragonnet (Shi et al., 2019) applies regularization to the entire network.

In summary, the notable aspects of our model are as follows: the introduction of a new latent variable $\mathbf{z}_o$ for factors unrelated to outcome and/or treatment to aid the recovery of the true underlying structure; the ability to estimate both individual level and average treatment effects; and, as far as we are aware, the first incorporation of targeted learning in a deep latent variable approach, and one which backpropagates the regularization gradients to specific outcome-related parameters in the network.

## 4 RELATED WORK

There are a number of ways to mitigate the problems associated with the confounding between the covariates and the treatment. For a review on such methods, readers are pointed to the recent survey by (Yao et al., 2020). Here we consider methods that utilize neural networks as part of their models, but note that many non-neural network methods exist (Chernozhukov et al., 2017; van der Laan & Rose, 2011; van der Laan & Starmans, 2014; Rubin, 2005; Hill, 2011; Athey & Imbens, 2016).

Perhaps the most similar works to ours are those of Dragonnet (Shi et al., 2019) and TEDVAE (Zhang et al., 2020). We discuss the differences between these and TVAE in turn. Dragonnet

incorporates the same targeted learning regularization process which allows for the simultaneous optimization of $Q$ and $\epsilon$. However, the method sacrifices the ability to estimate individual treatment effect in preference to achieving good estimation of the average treatment effect across the sample. Indeed, they do not report PEHE. Finally, Dragonnet applies regularization to the entire network, whereas we specifically 'target' the regularization to the outcome prediction arm by restricting the backpropagation of gradients.

TEDVAE, on the other hand, builds on CEVAE (Louizos et al., 2017) and seeks inference and disentanglement of the latent instrumental, risk, and confounding factors from proxy variables with a variational approach. However, it has no means to allocate latent variables that are unrelated to treatment and/or outcome (i.e., TVAE's $\mathbf{z}_o$). The advantage of including factors $\mathbf{z}_o$ with a variational penalty is that the model has the option to use them, or not to use them, depending on whether they are necessary (i.e. KL is pushed to zero). It is important not to force factors unrelated to treatment and outcome into $\mathbf{z}_{\{c,y,t\}}$ because doing so restricts the overlap between the class of models that can be represented using TEDVAE, and the class of models describing the true data generating process.

Other methods include GANITE (Yoon et al., 2018) which requires adversarial training, and may therefore be more difficult to optimise. PM (Schwab et al., 2019), SITE (Yao et al., 2018), and MultiMBNN (Sharma et al., 2020) incorporate propensity score matching. TARNET (Shalit et al., 2017) inspired the two-headed outcome arm in our TVAE, as well as the three-headed architecture in (Shi et al., 2019). RSB (Zhang et al., 2019) incorporates a regularization penalty, based on the Pearson Correlation Coefficient, intended to reduce the association between latent variables predictive of treatment assignment and those predictive of outcome.

## 5 EXPERIMENTS

We perform an ablation study, beginning with (a) TVAE (base) which is equivalent to TEDVAE (b) TVAE with $\mathbf{z}_o$, and (c) TVAE with both with $\mathbf{z}_o$ and targeted regularization $\xi$ during training. In order to fairly evaluate the benefits of introducing $\mathbf{z}_o$, we ensure that the total number of latent dimensions remains constant. We undertake the ablation study on a synthetic dataset which we call TVAEsynth, before comparing against methods on both the IHDP dataset (Hill, 2011; Gross, 1993) and the Jobs dataset (LaLonde, 1986; Smith & Todd, 2005; Dehejia & Wahba, 2002). In particular, the synthetic dataset was intentionally created such that not all covariates are exogenous and so that there exist some latent factors not related to outcome or treatment. Thus, we should expect a significant improvement in performance to occur with the introduction of $\mathbf{z}_o$, demonstrating the importance of incorporating inductive bias that closely matches the true structure of the data. Note that while these datasets vary in whether the outcome variable is continuous (IHDP, TVAESynth) or binary (Jobs), the treatment variable is always binary. Whilst it is possible to undertake Targeted Learning on continuous treatment effects, we leave this to future work.

For the IHDP dataset, we evaluate our network on the Average Treatment Effect estimation error (eATE), and the Precision in Estimation of Heretogeneous Effect (PEHE). As per (Louizos et al., 2017; Shalit et al., 2017; Yao et al., 2018), for the Jobs dataset (for which we have only partial effect supervision) we evaluate our network on the Average Treatment effect on the Treated error (eATT) to approximate the eATE, and the policy risk $R_{pol}$ to approximate the error on the CATE. See the appendix for definitions of these metrics. When estimating treatment effects, 100 samples are drawn for each set of input covariates $\mathbf{x}_i$. We compare against Dragonnet (Shi et al., 2019), TEDVAE (Zhang et al., 2020), CEVAE (Louizos et al., 2017), GANITE (Yoon et al., 2018), Targeted Maximum Likelihood Estimation (TMLE) (van der Laan & Rose, 2018), and TARNET + variants (Shalit et al., 2017). We provide results for both within sample and out-of-sample performance. It is worth noting that within sample and out-of-sample results are equally valid for treatment effect estimation, because the network is never supervised on treatment effect (Shi et al., 2019).

For model selection we follow Louizos et al. (2017) and Zhang et al. (2020) and use the minimum validation loss on the total objective function. Whilst some model selection heuristics exist that serve as surrogates for the eATE itself (e.g., see Hassanpour & Greiner 2019 or Athey & Imbens 2016) we take the same view as Zhang et al. (2020), namely that the development of our model 'should be self-sufficient and not rely on others'. Furthermore, we undertake minimal hyperparameter tuning for the simple reason that, in real-world applications, the supervision required for effective tuning would not be available. For all experiments, we undertake 100 replications and provide mean and

standard error. See the appendix for details on these datasets and the architecture, as well as training and testing details.

Table 1: Means and standard errors for the ablation study using the TVAESynth dataset. Here, 'oos' means out-of-sample, 'ws' means within sample. '$+\mathbf{z}_o$' indicates the introduction of the miscellaneous factors, '$+\mathbf{z}_o^*$' indicates the introduction of miscellaneous factors but *without* changing the dimensionality of $\mathbf{z}_c$ thereby increasing total latent capacity, '$+\xi$' indicates our targeted regularization (with selected backpropagation), '$+\xi^*$' indicatest targeted regularization equivalent to the one used by Shi et al. (2019) (i.e. with gradients applied to all upstream parameters), the $\xi$ subscript indicates its weight in the loss.

| Method | $\sqrt{\epsilon_{PEHE}}$ ws | $\sqrt{\epsilon_{PEHE}}$ oos | $\epsilon_{ATE}$ ws | $\epsilon_{ATE}$ oos |
|---|---|---|---|---|
| TVAE (base/TEDVAE) | .179±.003 | .178±.003 | .128±.005 | .128±.005 |
| TVAE + $\mathbf{z}_o^*$ | .174±.003 | .173±.003 | .121±.005 | .120±.005 |
| TVAE + $\mathbf{z}_o$ | .166±.003 | .166±.003 | .069±.004 | .069±.004 |
| TVAE + $\xi_{\lambda=0.1}$ | .171±.003 | .170±.003 | .122±.004 | .121±.004 |
| TVAE + $\mathbf{z}_o + \xi_{\lambda=0.1}^*$ | **.151±.002** | **.150±.003** | **.048±.003** | **.048±.003** |
| TVAE + $\mathbf{z}_o + \xi_{\lambda=0.1}$ (full model) | **.150±.002** | **.150±.003** | **.048±.003** | **.048±.003** |

*Ablation Study - TVAESynth:* The results for the ablation study on the synthetic dataset are shown in Table 1. The results demonstrate that both eATE and PEHE are significantly improved by the incorporation of $\mathbf{z}_o$ or targeted regularization, with a combination of the two yielding the best results for both within sample and out of sample testing. The fact that TVAE $+\mathbf{z}_o$ outperforms TVAE $+\mathbf{z}_o^*$ despite the latter having a larger latent capacity, suggests that reducing the capacity of the latent space has a beneficial, regularizing effect. Based on the results of this ablation, the benefits of this regularizing effect appear to be distinct from the benefits that derive from the addition of miscellaneous factors. Finally, the results indicate negligible empirical benefits to restricting the backgpropagation of the regularizer to non-propensity related parameters. However, the restriction of the backpropagation (according to our implementation), more closely aligns with the original TMLE and efficient influence curve theory, and we therefore retain this feature for the remaining experiments.

*IHDP Results:* Results on IHDP are shown in Table 2 and indicate state of the art performance for both within sample and out-of-sample eATE and PEHE. The results corroborate the ablation results in Table 1, in that the incorporation of $\mathbf{z}_o$ and targeted regularization result in monotonic improvement above TEDVAE. TVAE is outperformed only by Dragonnet on within-sample eATE performance. However, this method does not provide estimations for indvidual CATE, and is limited to the estimation of average treatment effects.

Table 2: Means and standard errors for evaluation on the semi-synthetic IHDP dataset (Hill, 2011). Results from: (Louizos et al., 2017; Shalit et al., 2017; Zhang et al., 2020; Yoon et al., 2018). Here, 'oos' means out-of-sample and 'ws' means within sample. '+ $\mathbf{z}_o$' indicates the introduction of the miscellaneous factors, and '+ $\xi$' indicates targeted regularization with subscript indicating its weight in the loss.

| Method | $\sqrt{\epsilon_{PEHE}}$ ws | $\sqrt{\epsilon_{PEHE}}$ oos | $\epsilon_{ATE}$ ws | $\epsilon_{ATE}$ oos |
|---|---|---|---|---|
| TMLE (van der Laan & Rose, 2018) | 5.0±.20 | - | .30±.01 | - |
| CEVAE (Louizos et al., 2017) | 2.7±.10 | 2.6±.10 | .34±.01 | .46±.020 |
| TARNet (Shalit et al., 2017) | .88±.00 | .95±.00 | .26±.01 | .28±.01 |
| CFR-MMD (Shalit et al., 2017) | .73±.00 | .78±.00 | .30±.01 | .31±.01 |
| CFR-Wass (Shalit et al., 2017) | .71±.00 | .76±.00 | .25±.01 | .27±.01 |
| TEDVAE (Zhang et al., 2020) | .62±.11 | .63±.12 | - | .20±.05 |
| GANITE (Yoon et al., 2018) | 1.9±.40 | 2.4±.40 | .43±.05 | .49±.05 |
| Dragonnet w/ t-reg (Shi et al., 2019) | - | - | **.14±.01** | .20±.01 |
| TVAE (w/ $\mathbf{z}_0, \xi_{\lambda=0.0}$) | **.57±.03** | **.57±.03** | **.16±.01** | **.16±.01** |
| TVAE (w/ $\mathbf{z}_0, \xi_{\lambda=0.4}$) | **.52±.02** | **.54±.02** | **.15±.01** | **.16±.01** |

Table 3: Means and standard errors for evaluation on the real-world Jobs dataset (LaLonde, 1986; Smith & Todd, 2005; Shalit et al., 2017; Yoon et al., 2018). Results taken from: (Louizos et al., 2017; Zhang et al., 2020). Here, 'oos' means out-of-sample and 'ws' means within sample. '+ $\mathbf{z}_o$' indicates the introduction of the miscellaneous factors, and '+ $\xi$' indicates targeted regularization with subscript indicating its weight in the loss.

| Method | $R_{pol}$ ws | $R_{pol}$ oos | $\epsilon_{ATT}$ ws | $\epsilon_{ATT}$ oos |
|---|---|---|---|---|
| TMLE (van der Laan & Rose, 2018) | .22±.00 | - | .02±.01 | - |
| CEVAE (Louizos et al., 2017) | .15±.00 | .26±.00 | .02±.01 | .03±.01 |
| TARNet (Shalit et al., 2017) | .17±.00 | .21±.00 | .05±.02 | .11±.04 |
| CFR-MMD (Shalit et al., 2017) | .18±.00 | .21±.00 | .04±.01 | .08±.03 |
| CFR-Wass (Shalit et al., 2017) | .17±.00 | .21±.00 | .04±.01 | .09±.03 |
| GANITE (Yoon et al., 2018) | **.13±.00** | **.14±.00** | **.01±.01** | .06±.03 |
| TEDVAE (Zhang et al., 2020) | - | - | .06±.00 | .06±.00 |
| TVAE (w/ $\mathbf{z}_0$, $\xi_{\lambda=0}$) | .16±.00 | .16±.00 | **.01±.00** | **.01±.00** |
| TVAE (w/ $\mathbf{z}_0$, $\xi_{\lambda=1}$) | .16±.00 | .16±.00 | **.01±.00** | **.01±.00** |

*Jobs Results:* The results for the Jobs data are shown in Table 3. GANITE was found to perform the best across most metrics, although this method has been argued to be more reliant on larger sample sizes than others, on the basis that it performs relatively poorly on the smaller IHDP dataset (Yoon et al., 2018). Furthermore, GANITE relies on potentially unstable/unreliable adversarial training (Moyer et al., 2018; Lezama, 2019; Gabbay & Hosen, 2019). Finally, TVAE outperforms GANITE on eATT, is consistent (beyond 2 decimal places) across out-of-sample and within-sample evaluations and has a lower standard err, and is competitive across all metrics. On this dataset, the concomitant improvements associated with the additional latent factors and targeted learning were negligible.

## 6 CONCLUSION

In this work we aimed to improve existing latent variable models for causal parameter estimation in two ways: Firstly, by introducing a latent variable to model factors unrelated to treatment and outcome, thereby enabling the model to more closely reflect the data structure; and secondly, by incorporating a targeted learning regularizer with selected backpropagation to further de-bias outcome predictions. Our experiments demonstrated concomitant improvements in performance, and our comparison against other methods demonstrated TVAE's ability to compete with and/or exceed state of the art for both individual as well as average treatment effect estimation. For future work, we plan to explore the application of TVAE to longitudinal data with continuous or categorical treatment. There is also opportunity to explore the use of TVAE in inferring hidden confounders from proxies in the dataset, as well as interpreting the model to explore and validate what information is inferred in the model's latent factors. Additionally, it was noted from the ablation study results that the restriction of regularization gradients did not yield a significant change in performance when compared with applying the regularization to the entire network. As well as undertaking further experiments to understand this behavior, we propose to explore alternative ways to integrate the targeted learning update procedure to the learning procedure. Finally, the ablation results indicated that part of the improvement associated with the introduction of $\mathbf{z}_o$ is associated with a regularizing effect relating to the reduction in the dimensionality of $\mathbf{z}_c$. This aspect of the model's behavior also lends itself to future exploration.

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

# A   APPENDIX

This appendix includes details on the metrics used for evaluating treatment effect estimation; background on Variational AutoEncoders; a formulation of the targeted learning regularizer for continuous, unbounded outcomes (as used with the IHDP dataset evaluations); details about the datasets; and training, testing, hyperparameters, architecture, and hardware details. Source code and data (including IHDP (Hill, 2011; Gross, 1993), Jobs (LaLonde, 1986; Smith & Todd, 2005), and our own TVAESynth) will also be attached as supplementary material for reproducibility purposes upon publication.

## A.1   METRICS

This section presents the metrics used for the experiments (Section 5 in the main paper).

The Average Treatment Effect (ATE) and error on the estimation of ATE (eATE) is given in Eq. 8.

$$\hat{\boldsymbol{\tau}}(Q; \mathbf{x}) = \frac{1}{N} \sum_{i=1}^{N} (Q(1, \mathbf{x}_i) - Q(0, \mathbf{x}_i)), \quad \epsilon_{ATE} = |\frac{1}{N} \sum_{i=1}^{N} (\hat{\boldsymbol{\tau}}(Q; \mathbf{x}_i) - \tau(\mathbf{x}))| \tag{8}$$

To estimate the error on the model's capacity to model the Conditional Average Treatment effect (CATE), the Precision in Estimation of Heteogenous Effects (PEHE) is given in Eq. 9.

$$\epsilon_{PEHE} = \sqrt{\frac{1}{N} \sum_{i=1}^{N} (\hat{\boldsymbol{\tau}}(Q; \mathbf{x}_i) - \tau(\mathbf{x}_i))^2} \tag{9}$$

It can be seen from Equations 8 that the ATE is essentially the expectation of the conditional treatment effect (conditioned on the covariates for each individual) over the data set (Jesson et al., 2020). For scenarios when a proportion of the dataset is from a Randomized Controlled Trial (RCT), as is the case for the Jobs dataset, we may use the error on the estimation of Average Treatment effect on the Treated (ATT), which is given in Eq. 10 (Shalit et al., 2017; Louizos et al., 2017):

$$eATT = |\frac{1}{|T_1|} \sum_{i \in T_1} y_i - \frac{1}{|T_0|} \sum_{j \in T_0} y_j - \frac{1}{|T_1|} \sum_{i \in T_1} (Q(1, \mathbf{x}_i) - Q(0, \mathbf{x}_i))| \tag{10}$$

where $T = T_1 \cup T_0$ constitutes all individuals in the RCT, and the subscripts denote whether or not those individuals were in the treatment (subscript 1) or control groups (subscript 0). The first two terms in Eq. 10 comprise the true ATT, and the third term the estimated ATT. In datasets where the ground-truth for the CATE is not available (as is the case with the Jobs dataset) we may use the policy risk as a proxy for PEHE:

$$\mathcal{R}_{pol} = 1 - \left( \mathbb{E}[\mathbf{y}^{t=1}|\pi(\mathbf{x}) = 1]p(\pi(\mathbf{x}) = 1) + \mathbb{E}[\mathbf{y}^{t=0}|\pi(\mathbf{x}) = 0]p(\pi(\mathbf{x}) = 0) \right) \tag{11}$$

where $\pi(\mathbf{x}_i) = 1$ is the policy to treat when $\hat{y}_i^{t=1} - \hat{y}_i^{t=0} > \epsilon$, and $\pi(\mathbf{x}_i) = 0$ is the policy not to treat otherwise (Yao et al., 2018; Shalit et al., 2017). $\epsilon$ is a treatment threshold. This threshold can be varied to understand how treatment inclusion rates affect the policy risk. For our experiments we set $\epsilon$ to zero, as per (Shalit et al., 2017; Louizos et al., 2017).

## A.2  VARIATIONAL AUTOENCODERS

This part of the appendix is referenced in Section 3 (Methodology) of the main paper.

We now consider the theory behind a popular and powerful latent variable representation learning and density estimation method known as Variational AutoEncoders (VAEs) (Kingma & Welling, 2014). Although adversarial methods (Goodfellow et al., 2014) have been shown to be effective for density estimation, they are also troublesome to train. Hence we choose VAEs which have more stable training dynamics (Moyer et al., 2018). A further motivation for the use of VAEs relates to their ability to infer latent variables. The Ignorability assumption holds that all confounders are observed, and much causal inference is undertaken with this assumption (Mooij et al., 2010; Maier et al., 2013; Oktay, 2018; Silva et al., 2006). The use of a latent variable model allows us to infer unobserved/hidden confounders (Louizos et al., 2017), although the use of VAEs means that there is some uncertainty concerning the guarantees of the learned model (Rolinek et al., 2019; Dai et al., 2019; Lucas et al., 2019a;b). This uncertainty notwithstanding, previous implementations for latent variable models with causal inference show promising results in comparison with other methods (Louizos et al., 2017; Mayer et al., 2020; Hassanpour & Greiner, 2020).

We wish to encode the $m$-dimensional covariates $\mathbf{x}$ on a manifold via a stochastic mapping $p(\mathbf{z}|\mathbf{x})$, where latent variables $\mathbf{z}$ provide a compact representation. The distribution $p_\theta(\mathbf{z}|\mathbf{x})$ also serves as the posterior to the generative model $p_\theta(\mathbf{x}) = \int p_\theta(\mathbf{x}|\mathbf{z})p(\mathbf{z})d\mathbf{z}$ having parameters $\theta$. Marginalizing out $\mathbf{z}$ is usually intractable, and the true posterior $p(\mathbf{z}|\mathbf{x})$ is unknown, so a simpler approximating posterior $q_\phi(\mathbf{z}|\mathbf{x})$ having parameters $\phi$ is introduced (Blei et al., 2018) such that:

$$\log p_\theta(\mathbf{x}) = \mathbb{E}_{q_\phi(\mathbf{z}|\mathbf{x})}\left[\log \frac{p_\theta(\mathbf{x}|\mathbf{z})p(\mathbf{z})}{q_\phi(\mathbf{z}|\mathbf{x})} + \log \frac{q_\phi(\mathbf{z}|\mathbf{x})}{p_\theta(\mathbf{z}|\mathbf{x})}\right] \tag{12}$$

Taken together with the expectation, the second term on the right hand side Eq. 12 represents the Kullback-Liebler Divergence (KLD) between the approximating posterior and the true posterior. The KLD is always positive, and by ignoring this term, we are left with a lower-bound on the log-likelihood known as the variational lower bound, or Evidence Lower BOund (ELBO). The VAE provides the means to scale this amortized variational inference to intractable, high-dimensional problems, and minimizes the negative log likelihood over a dataset of $N$ samples by adjusting the parameters of neural networks $\{\theta, \phi\}$ according to the ELBO.

$$\frac{1}{N}\sum_{i=1}^{n} -\log p_\theta(\mathbf{x}_i) \leq \frac{1}{N}\sum_{i=1}^{N}\left(-\mathbb{E}_{q_\phi(\mathbf{z}|\mathbf{x}_i)}\left[\log p_\theta\left(\mathbf{x}_i|\mathbf{z}\right)\right] + \beta\mathbb{D}_{KL}\left[q_\phi(\mathbf{z}|\mathbf{x}_i)\|p(\mathbf{z})\right]\right), \tag{13}$$

where $\beta = 1$ is used for the standard variational approximation procedure, but may be set empirically (Higgins et al., 2017), annealed (Burgess et al., 2018) or optimized according to the Information Bottleneck principle (Alemi et al., 2017; Tishby & Zaslavsky, 2015). The first term in Eq. 13 is the negative log-likelihood and is calculated in the form of a reconstruction error. The second term is the KLD between the approximating posterior and the prior, and therefore acts as a prior regularizer. Typically, the family of isotropic Gaussian distributions is chosen for the posterior $q_\phi(.)$, and an isotropic Gaussian with unit variance for the prior $p(\mathbf{z})$, which helps to encourage disentanglement (Higgins et al., 2017).

## A.3  TARGETED REGULARIZATION FOR BOUNDED AND UNBOUNDED OUTCOMES

This part of the appendix is referenced in Section 3 (Methodology) of the main paper. In contrast with Eq. 6 in the main paper, this formulation of targeted regularization is for an unbounded, continuous outcome (as is the case in the IHDP experiments). A mean-squared error version of $\xi$, similar to the one found in (Shi et al., 2019), is given as follows:

$$\hat{Q}(t_i, \mathbf{z}_i^y, \mathbf{z}_i^c, \epsilon) = Q(t_i, \mathbf{z}_i^y, \mathbf{z}_i^c) + \epsilon\left(\frac{I(t_i = 1)}{g(t_i = 1; \mathbf{z}_i^t, \mathbf{z}_i^c)} - \frac{I(t_i = 0)}{g(t_i = 0; \mathbf{z}_i^t, \mathbf{z}_i^c)}\right) \tag{14}$$

$$\xi_i(\hat{Q}, g; \phi_{c,t,y}, \epsilon) = (y_i - Q(t_i, \mathbf{z}_i^y, \mathbf{z}_i^c, \epsilon))^2 \tag{15}$$

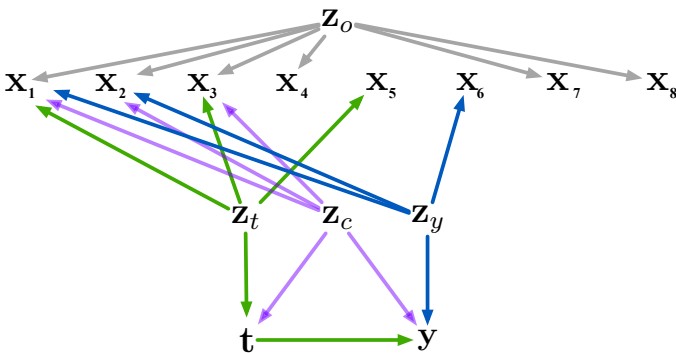

Figure 4: The DAG for TVAESynth dataset.

However, we stress that this formulation is only appropriate for unbounded outcomes, and not continuous outcomes in general (van der Laan & Rose, 2011). For continuous, bounded outcomes, as well as binary outcomes, the NLL formulation (as in the main paper) has been shown to be more appropriate, and theoretically sound.

### A.4 DATASETS

This part of the appendix is referenced in Section 5 (Experiments) of the main paper.

We utilize 100 replications of the semi-synthetic *Infant Health and Development Program (IHDP)* dataset (Hill, 2011; Gross, 1993)[4] The linked version (see footnote) corresponds with setting A of the NPCI data generating package (Dorie, 2016), which is the version that is used most frequently in other comparisons (e.g., see Shi et al. 2019; Shalit et al. 2017; Yao et al. 2018) and comprises 608 untreated and 139 treated samples (747 in total). There are 25 covariates, 19 of which are discrete/binary, and the rest are continuous. The outcome for the IHDP data is continuous and unbounded. Similarly to (Louizos et al., 2017; Shalit et al., 2017) and others, we utilize a 60/30/10 train/validation/test split.

We also utilize the job outcomes dataset which we refer to as *Jobs* (LaLonde, 1986; Smith & Todd, 2005).[5] Unlike the IHDP datset, Jobs is real-world data with a binary outcome. We follow a similar procedure to (Shalit et al., 2017) who indicate that they used the Dehejia and Wahba (Dehejia & Wahba, 2002) and PSID comaprison samples. The Dehejia-Wahba sample comprises 260 treated samples and 185 control samples, along with the PSID comparison group comprising 2490 samples'. The dataset contains a mixture of observational and Randomized Controlled Trial data. Similarly to (Louizos et al., 2017; Shalit et al., 2017) and others, we utilize a 56/24/20 train/validation/test split, and undertake 100 runs with varying random split allocations in order to acquire an estimate of average performance and standard error. Note that, between models, the same random seed is used both for intialization as well as dataset splitting, and therefore the variance due to these factors is equivalent across experiments.

Finally, we introduce a new synthetic dataset named *TVAESynth* which follows the structure shown in Figure 4. While the weightings are chosen relatively arbitrarily, the structure is intentionally designed such that there are a mixture of exogenous and endogenous covariates. This enables us to compare the performance of TVAE with and without $z_o$ (while keeping the total number of latent dimensions constant). These data comprised 1000 samples, a continuous outcome and binary treatment, 8 covariates, and generative/structural equations as follows:

$$\mathbf{U}_{z_o,z_c,z_t,z_y,y} \sim \mathcal{N}(\mathbf{0},\mathbf{1}) \quad \mathbf{U}_{x_1,x_4,t} \sim \text{Bernoulli}(0.5) \quad \mathbf{U}_{x_{2:3},x_{5:8}} \sim \mathcal{N}(\mathbf{0},\mathbf{1}) \quad (16)$$

---

[4]Available from `https://www.fredjo.com/`, `https://github.com/WeijiaZhang24/TEDVAE/` and elsewhere, and will be included in supplementary folder with source code upon acceptance.

[5]Available from `https://users.nber.org/~rdehejia/data/.nswdata2.html`.

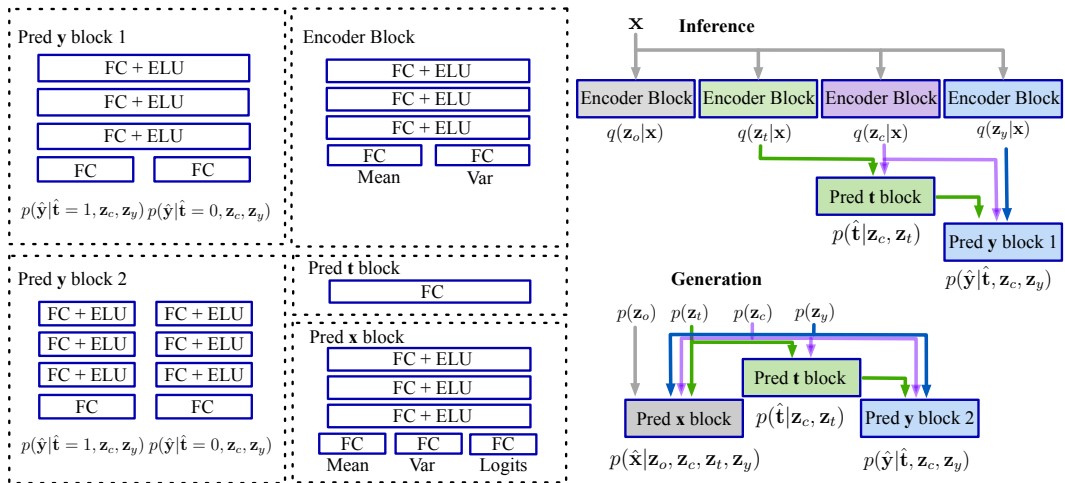

Figure 5: Block architectural diagram for TVAE's inference and generation models (number of layers in Pred **y** 1 & 2 and Pred **x** blocks varies by experiment, as does the number of neurons in each layer (see details in text).

$$\mathbf{z}_o = \mathbf{U}_{z_o} \quad \mathbf{z}_y = \mathbf{U}_{z_y} \quad \mathbf{z}_t = \mathbf{U}_{z_t} \quad \mathbf{z}_c = \mathbf{U}_{z_c} \tag{17}$$

$$\mathbf{x}_1 \sim \text{Bernoulli}(\sigma(\mathbf{z}_t + 0.1(\mathbf{U}_{x_1} - 0.5))) \quad \mathbf{x}_2 \sim \mathcal{N}(0.4\mathbf{z}_o + 0.3\mathbf{z}_c + 0.5\mathbf{z}_y + 0.1\mathbf{U}_{x_2}, 0.2) \tag{18}$$

$$\mathbf{x}_3 \sim \mathcal{N}(0.2\mathbf{z}_o + 0.2\mathbf{z}_c + 1.2\mathbf{z}_t + 0.1\mathbf{U}_{x_3}, 0.2) \quad \mathbf{x}_4 \sim \text{Bernoulli}(\sigma(0.6\mathbf{z}_o + 0.1(\mathbf{U}_{x_4} - 0.5))) \tag{19}$$

$$\mathbf{x}_5 \sim \mathcal{N}(0.6\mathbf{z}_t + 0.1\mathbf{U}_{x_5}, 0.1) \quad \mathbf{x}_6 \sim \mathcal{N}(0.9\mathbf{z}_y + 0.1\mathbf{U}_{x_6}, 0.1) \tag{20}$$

$$\mathbf{x}_7 \sim \mathcal{N}(0.5\mathbf{z}_o + 0.1\mathbf{U}_{x_7}, 0.1) \quad \mathbf{x}_8 \sim \mathcal{N}(0.5\mathbf{z}_o + 0.1\mathbf{U}_{x_8}, 0.1) \tag{21}$$

$$\mathbf{t}_p = \sigma(0.2\mathbf{z}_c + 0.8\mathbf{z}_t + 0.1\mathbf{U}_t) \quad \mathbf{t} \sim \text{Bernoulli}(\mathbf{t}_p) \tag{22}$$

$$y := 0.2\mathbf{z}_c + 0.5\mathbf{z}_y\mathbf{t} + 0.2\mathbf{t} + 0.1\mathbf{U_y} \tag{23}$$

Interventional distributions were generated by setting **t** equal to 1 and 0 thereby yielding ground truth ATE ($\approx 0.8$) and individual effects for evaluation purposes. The number of treated individuals is $\approx 70\%$. Note that for every replication, the 1000 sample evaluation set is regenerated, and so may have an empirical statistics that vary. For experiments, we use a 80/20 train/test split.

## A.5 MODEL AND HYPERPARAMETERS

This part of the appendix is referenced in Section 3 (Methodology), as well as in Section 5 (Experiments) of the main paper. A block diagram of the TVAE architecture is shown in Fig. 5. For continuous outcomes (as in the IHDP dataset) we standardize the values and model as a Gaussian with a fixed variance of 1, and a mean determined by the outcome arm. All binary outcomes in the model (e.g. treatment or the relevant covariates) are modelled as Bernoulli distributed with a probability determined by the associated neural network function.

We now list the hyperparameters that were explored as part of model training. There may be room to improve on our figures with further hyperparameter tuning. However, given that the tuning of

hyperparameters in a causal inference paradigm is problematic in general, we intentionally limited the space of hyperparameters (similarly to Zhang et al. 2020). Bold font indicates the settings that were used in the presented results.

Hyperparameter settings for IHDP dataset experiments were: hidden layers: **3**; the weight on targeted regularization $\lambda_{TL} = \{0.0, 0.1, 0.2, \mathbf{0.4}, 0.6, 0.8, 1.0\}$; an **Adam** (Kingma & Ba, 2017) optimizer with learning rate $LR = \{$1e-3, 1e-4, **5e-5**$\}$; number of hidden neurons was **300**; number of layers = **4**; dimensionality of the latent factors was $D_{z_t} = D_{z_t} = \mathbf{10}, D_{z_c} = \mathbf{15}, D_{z_o} = \mathbf{5}$; batch size of **200**; number of epochs **200**; weight regularization **1e-4**; and learning rate decay **5e-4**.

Hyperparameter settings for the Jobs dataset experiments were: hidden layers: **3**; the weight on targeted regularization $\lambda_{TL} = \{0.0, \mathbf{0.1}, 0.2, 0.4, 0.6, 0.8, 1.0\}$; an **Adam** (Kingma & Ba, 2017) optimizer with learning rate $LR = \{$5e-5, **1e-5**$\}$; number of hidden neurons was **200**; number of layers = **2**; dimensionality of the latent factors was $D_{z_t} = D_{z_t} = \mathbf{6}, D_{z_c} = \mathbf{8}, D_{z_o} = \mathbf{4}$; batch size of **200**; number of epochs **200**; weight regularization **1e-4**; and learning rate decay **5e-4=3**.

Hyperparameter settings for the TVAESynth dataset experiments were: hidden layers: **2**; the weight on targeted regularization $\lambda_{TL} = \{0.0, \mathbf{0.1}, 0.2, 0.4, 0.6, 0.8, 1.0\}$; an **Adam** (Kingma & Ba, 2017) optimizer with learning rate $LR = $ **5e-5**; number of hidden neurons was **20**; number of layers = **2**; dimensionality of the latent factors was $D_{z_t} = D_{z_t} = D_{z_c} = \mathbf{2}, D_{z_o} = \mathbf{1}$; batch size of **200**; number of epochs **40**; weight regularization **1e-4**; and learning rate decay **5e-3**.

As described in the main paper, wherever a range of hyperaparameters was explored, the validation loss on the total objective function was used as the model selection criterion (i.e., not the causal effect estimation performance, which is not available in real-world scenarios).

## A.6  SOFTWARE AND HARDWARE

The network is coded using Pyro (Bingham et al., 2019) and builds on base code by (Zhang et al., 2020). We train on a GPU (e.g. NVIDIA 2080Ti) driven by a 3.6GHz Intel I9-9900K CPU running Ubuntu 18.04. Training 200 epochs of the IHDP dataset (training split of 450 samples) takes approx. 35 seconds (0.175s per epoch).

