# OpenReview forum: "Targeted VAE: Structured Inference and Targeted Learning for Causal Parameter Estimation"
_ICLR.cc/2021/Conference — Reject_

### Official Review · AnonReviewer4 · 2020-10-25
**Proposes improvements that lead to increase in performance over SoTA**

**Rating:** 6
**Confidence:** 4

**Review:**

The paper proposes Targeted Variational Autoencoders for ATE and CATE estimation.

It is assumed that all relevant confounding variables can be measured through proxy variables and then a VAE architecture is used to disentangle the latent variables into 4 sets: z_t - confounding between the treatment and covariates, z_c - confounding between the treatment, covariates and target, z_y - confounding between the covariates and target, z_0 - solely related to the covariates.

The proposed architecture builds on an existing approach, TEDVAE. The main difference is that TEDVAE did not include the z_0 term.

A targeted regularization approach is also used which is similar to Shi et al. 2019

The proposed method outperforms existing approaches on to real world datasets.

In general the paper is well written and the approach appears sound and reasonable. The novelty is somewhat limited as both the architecture and regularization build incrementally on previous approaches. However, the proposed method appears to lead to a decent increase in performance over SoTA approaches.

---

> ### Author Response · Authors · 2020-11-17
> **Response to AnonReviewer4**
>
> We thank AnonReviewer4 for their recognition of the performance of our method, that the paper is 'well written', and for their positive review.
>
> We understand the reviewer's concerns regarding novelty. We would reason that (to use the words of  AnonReviewer3) TVAE is intended to make a "complete picture" by updating and bringing together latent variable models and targeted learning for causal inference into one model. Furthermore, Targeted Learning (broadly speaking) is in our view an underappreciated technique in the causal inference-ML community, which is worthy of exploration and integration into modeling techniques.. Indeed, we are excited about its potential in the field.
>
> We have made a number of changes to the paper according to the constructive feedback from all reviewers, and we hope that by addressing the feedback the work has been improved further. We have revised and uploaded an updated version of the manuscript accordingly, regarding which we very much welcome further comments.

---

### Official Review · AnonReviewer2 · 2020-10-28
**Not yet ready for publication**

**Rating:** 3
**Confidence:** 5

**Review:**

This paper proposes a method for estimating conditional and average treatment effects under unconfoundedness. There are two main ideas: (1) train a VAE with latent space aimed at the adjustment-relevant information, and (2) incorporate the targeted regularization of Shi et al into the training.

The problem of estimating causal effects using deep learning is important, and I think paper has a promising direction. It is closely related to a growing literature on this subject (as noted in the paper itself), and does a reasonable job of explaining the innovations.

However, there are some significant issues.

The exposition is generally unclear. The paper needs a complete rewrite, with particular attention to clarity about the (salient parts of) TMLE, and the claimed advantages of this approach over closely related methods. There also needs to be substantial improvement around the VAE component of the model, and precisely what the identification assumptions are.

The empirical results are inadequate. The IHDP and LaLonde datasets are not challenging enough to separate methods in the bakeoff, nor rich enough to give substantial insight into their performance. I suggest you use ACIC competition data. Further, the ablation study doesn't seem to test the right aspects of the model. E.g., there should be some experiment showing that not applying targeted regularization to the propensity score part of the model leads to better performance, since that's the core distinction with Shi et al.

This paper isn't ready for publication in its current form. However, I reiterate that I think the basic ideas are interesting, and worth developing.

Some further comments and questions:

1. missing expectation symbol in 2nd paragraph of background (in def of \tau)
2. equation 1 is meant to be describing an observational quantity; the do(T=t) should just be t
3. the discussion about optimal epsilon=0 is confusing (I understand that you mean that running more than 1 round of the TMLE update doesn't make a difference, but this is not clear in your writing)
4. indeed, all of the prose around equation 2 should be rewritten. It is not clear in the current version.
5. under DAG (c), x doesn't block backdoor paths and strong ignorability is not satisfied
6. the related work should be clearer that targeted regularization was proposed in Shi et al
7. what models are you using for Q and g for the TMLE reported in the table? In particular, are you comparing training with targeted regularization to training without targeted regularization and then plugging in to TMLE?
8. when you compare to 'Dragonnet' are you comparing to Dragonnet + Targeted Regularization, or just vanilla dragonnet?
9. what exactly is done to prevent the propensity score from being modified by the targeted regularization? I presume that at least the shared latent representation is still affected by the inclusion of the targeted regularization term?

---

> ### Author Response · Authors · 2020-11-17
> **Response to AnonReviewer2**
>
> We thank AnonReviewer2 for their constructive feedback, and their acknowledgement that the 'ideas are interesting'. We seek to address their concerns in the same order in which they were presented. Based on your initial feedback we have revised and uploaded an updated version of the manuscript, regarding which we are grateful for further feedback.
>
> Regarding improving the clarity of the important aspects and advantages of TMLE, we have sought to improve this in the new version of the paper. We have also expanded our justifications for using a VAE, and have elaborated on the identification/assumptions [penultimate para p.4, 2nd para p.5]. We politely and respectfully disagree that the paper needs a 'complete rewrite', and tentatively note that the other reviewers have not come to the same conclusion. We also note the reviewer's own comment that we do 'a reasonable job of explaining the innovations', and hazard that the reviewer does not believe that these particular parts need rewritten? That being said, we do wish to improve the paper wherever possible, and are very grateful for any additional and specific suggestions for where to make such improvements.
>
> We thank AnonReviewer2 for their suggestion regarding additional experiments. As well as adding more ablation study results (also see below), we are current adapting the code to run the ACIC2016 competition data in order to compare against TEDVAE on a richer dataset. The dataset is relatively large, and we wish to undertake 10 replications of the 77 settings (following TEDVAE's procedure) which takes some time to run. In the meantime, please see Table 1 in the TEDVAE paper (https://arxiv.org/pdf/2001.10652.pdf). Given that our model builds upon TEDVAE, we expect to match or exceed their performance, which is considerably better than the CEVAE, SITE, CFR, and competitive with X-RF methods.
>
> Regarding the 'core distinction with Shi', we wish to highlight that our model is considerably different to Dragonnet overall - TVAE is a generative, probabilistic, latent variable model, and Dragonnet is not. This notwithstanding, we do acknowledge the similarity between the targeted regularization component. Our primary motivation for limiting the backpropagation of the gradients from the targeted regularization is theoretic and, as AnonReview1 noted, such an implementation is closer to the original targeted learning formulation. We have made this clearer in the paper [see penultimate paragraph in section 3]. For completeness, we have also re-run the ablation study (as the reviewer suggested) to check the impact of allowing gradients to be applied to the propensity arm as well as the outcome arm, and our results indicate that applying generic TL reg to all of the network yields a negligible decrease in performance (i.e. the results are extremely close and  within the noise tolerance). We have added these results to the paper [see Table 1], but they suggest that the preference for closer theoretic consistency may not substantially manifest empirically. This notwithstanding, and given the close empirical results, we stand by our implementation difference.
>
> 1. We are not sure there is a missing expectation symbol for \tau_i as the reviewer suggests? This quantity is the expected treatment effect for a single individual -perhaps this is where the confusion has arisen? i.e. the CATE for individual 'i' is tau_i = E((y1_i - y0_i)|x_i)
> 2. Thanks, we have corrected this and made Eq.1 an observational quantity.
> 3. Agreed, we have clarified the paragraph about the TMLE update process [see paragraphs before and after Eq. 1]
> 4. We have also reworded the prose around Eq. 2.
> 5. Please could the reviewer clarify this point? We are not aware that we suggest that, under the DAG in Figure 1(c), x would block backdoor paths?
> 6.  We have updated the 2nd paragraph in the related work section to make this clearer.
> 7. The results for TMLE are taken from CEVAE and are also reproduced in Shalit et al. (2017). We have emailed the authors and will clarify upon response. We speculate that these were produced using the TMLE package for R, which incorporates a flexible ensemble of machine learning algorithms called a SuperLearner (see van der Laan & Rose, 2011 for more details).
> 8. We are comparing against Draggonnet with TL-reg, we have clarified this in Table 1.
> 9. To prevent the propensity score being modified we use the pytorch 'detach' method on the predicted propensity scores when they are used in the targeted regularization calculation. This prevents the propensity arm (and any upstream parameters) from being affected by the regularization, whilst allowing the same parameters to be affected by other influences (e.g. propensity classification likelihood loss). Indeed, the shared latent space *is* affected by targeted regularization, because it feeds the outcome arm, to which we apply the regularization. We have clarified this in the paper [see penultimate paragraph in Section 3].

---

### Official Review · AnonReviewer1 · 2020-10-28
**Nice addition to the causal nnet literature.**

**Rating:** 6
**Confidence:** 4

**Review:**

The proposed contribution of this work is to build of the existing literature which uses variational autoencoders for causal inference by (1) allowing an explicit mechanism for modeling irrelevant covariates and (2) incorporating targeted regularization into the latent variable nnet framework. Optimization of the model is done by minimizing the ELBO subject to a penalty term which the authors refer to as “targeted regularization”. This term is essentially an application of the TMLE model. Both of these proposals appear to improve the performance on what are now pretty standard benchmark datasets (jobs and ihdp).

Overall, I think this work presents two very sensible additions to the latent variable formulation of causal neural networks. My largest concern is with novelty–each contribution borders on incremental, and neither necessarily open doors for substantial amounts of follow on work. With that being said, I think that this work does provide value to the community given (a) the sensible, simple, model changes proposed and, (b) the pretty compelling empirical evidence.

Comments / Questions:

* You note that the gradients are taken with respect to zeta and not with respect to g_p or g_q in the paper. This language is kind of confusing at first read. As I read the difference is that you are more explicitly reproducing the machinery of TMLE by adding what is essentially a logistic regression (for bounded outcomes). This is very sensible, but would benefit from more explicit discussion relating back to the TMLE literature.
* It appears that the change mentioned above does not provide a substantial benefit for estimating the ATE over Dragonnet, can the authors provide intuition around that?
* The oblation study to tease apart the contribution of the two proposed changes is very interesting. Given that the introduction of zeta provides substantial benefit on its own, it would be interesting to see a variant that does not use z_0 but uses zeta.

Small edit notes:

* In the main text you make reference to equation 14 which is in the supplement. I believe you meant to make reference to equation 5.

---

> ### Author Response · Authors · 2020-11-17
> **Response to AnonReviewer1**
>
> We thank AnonReviewer1 for their recognition of the contributions in this work. We respond to their comments in the order in which they were presented by the reviewer. Based on your initial feedback we have revised and uploaded an updated version of the manuscript, regarding which we are grateful for further feedback.
>
> Regarding the reviewer's suggestion that our method limits the scope for further work, we have updated the conclusion to include some more open directions, some of which have arisen as part of the additional ablation study results. We also welcome any further clarity about why the reviewer felt that our work did not lend itself to future exploration.
>
> 1. We thank the reviewer for drawing attention to the fact that our implementation of the regularizer is, indeed, closer to the original targeted learning formulation, and we appreciate that this can be more clearly explicated. We have updated the wording accordingly [see penultimate paragraph in section 3].
> 2. It is difficult to identify the key reasons for the differences in performance - independently of whether targeted regularization is employed, or not, their model is fundamentally different from TVAE - and we agree this is an interesting open question. However, it is worth noting that Dragonnet does not provide CATE estimation (and indeed, they only provide eATE), so it might not be altogether surprising that a method that focuses just on a low eATE does better than a contemporary method that aims to achieve *both* low eATE and low eCATE [we also refer to this point in the 2nd paragraph of Section 4]
> 3. We thank the review for this suggestion, and have re-run the ablation to ensure consistency and added the suggested results to the ablation results table (Table 1). We have also added a new experiment to explore whether maintaining a constant total latent dimensionality after the introduction of $z_o$ had a regularizing effect on the network, and we welcome AnonReviewer1 to review the changes to Table 1 and the ablation study discussion.
> 4. The equation reference has been amended, thanks.

---

### Official Review · AnonReviewer3 · 2020-10-28
**Issues with the role of VAE and with the presentation**

**Rating:** 5
**Confidence:** 4

**Review:**

This work provides an improved method for individual and average causal parameter estimation. The main idea is to apply targeted learning to deep latent variable models. The proposed approach is based on two existing work:
- Disentangled variational latent model [Zhang et al., 2020].
- Implementing targeted learning in neural nets via regularization [Shi et al., 2019].

Considering the above two papers, the contribution of this work is not very significant. However, it provides a nice combination of those works and adds the missing element of z_0 to model factors unrelated to treatment and outcome and to give more flexibility to the design of the latent variables, which all together makes a complete picture. The authors compare the proposed method with some alternatives on two datasets and the results show the better performance of the approach in most of the cases.

In general, the presentation of the paper is subpar. In many parts only a high level explanation is provided. For instance,
- A motivation for using latent variable models is needed, similar to the one given in [Zhang et al., 2020].
- I do not believe that the explanation for equation (1) will clarify the matter.
- "The combination of KL regularization and supervision helps prevent z_o from learning information in z_{t,c,y}" This is an important point, yet it is very vaguely stated and should be elaborated.
- The last paragraph of page 5 regarding the difference between the training in this work and [Shi et al., 2019] is not clear.

On page 4, the authors mention that: "the use of deep latent variable techniques enables us to attempt to infer these hidden confounders from what are known as noisy proxy variables present in the dataset" This is in general not true. We are not able to infer the latent confounders, unless if we assume very strong assumptions on them. The aforementioned sentence is also confusing because in this paper it is assumed that we do not have any latent confounders, i.e., the ignorability assumption. If the authors are indeed assuming that we have latent confounders and they are estimated using VAE, then nothing can be said about the correctness of the outputs of the method. Therefore, the sentence "Doing so enables us to infer hidden confounders from proxy variables in the dataset, and to estimate the expected treatment effects, as well as individual-level treatment effects" in the introduction should also be clarified. The only way that the approach is justifiable is by assuming ignobility and assuming that VAE is only used to get a concise representation of the observed confounders to improve the prediction.

---

> ### Author Response · Authors · 2020-11-17
> **Response to AnonReviewer3**
>
> We thank AnonReview3 for recognizing the value in the work. Based on your feedback we have revised and uploaded an updated version of the manuscript, regarding which we welcome further comments. We seek to address the initial points raised and cover them in the same order in which the reviewer presented them:
>
> 1.  We have included a clearer motivation for the use of VAEs as suggested [see TVAE subsection in section 3]. In short, VAEs are needed if we are to infer disentangled latent variables which explain our observations, without restricting the functional form that maps between covariate space and latent space. By additionally structuring and disentangling this non-linearly inferred latent space (according to the TVAE DAG shown in Figure 1), we are able to improve estimation.
> 2. We have clarified the explanation of Eq. 1 [see paragraphs before and after Eq. 1 in updated paper].
> 3. This point has not been empirically tested and relates to the information bottleneck interpretation of VAEs and the rate-distortion tradeoff. In our view, it is actually a side-point, because whether or not there is overlap between what z_o and {z_y, z_c, z_t} does not affect the causal effect estimation (they are d-separated). Furthermore, the incorporation of z_o does not direct relevant information away from {z_y, z_c, z_t} because {z_y, z_c, z_t} are constrained to be predictive of t and y by the structure of the model. Given this, we tentatively propose that we remove the statement from the paper (to keep the exposition laconic), but if the reviewer still considers this issue to be a general concern that ought to be discussed, we are happy to add a paragraph on the topic. If the reviewer is interested, the comment in the paper originally arose due to a parallel thread relating to work on information bottleneck and the rate-distortion tradeoffs in VAEs. The combination of the KL pressure for the posterior to match an isotropic prior constrains the information capacity of the VAE latent space (for more information see e.g. Alemi et al. 2018).
> 4. We have clarified the differences between our implementation of targeted regularization and Dragonnet's [see penultimate paragraph in section 3]. In short, and as AnonReview1 has noted, our implementation aligns more closely with the theory of targeted learning, by restricting the influence of the regularizer to the outcome prediction arm of the network.
>
> Assumptions and Proxy Variables:
> Firstly, we understand that with observational data it is impossible to know (a) how many unobserved confounders there are, (b) whether there are sufficient proxies in the dataset from which the unobserved confounders might be inferred using a latent variable model, and (c) that, to borrow the reviewer's words, we cannot infer the latent confounders without 'some very strong assumptions on them'. Furthermore, the degree to which these issues are problematic will be dataset dependent.
> Secondly, we agree that it is confusing that we state that we assume ignorability, but then shift this assumption from 'all confounders are observed', to 'all unobserved confounders have been inferred from proxies'. We propose to clarify our assumptions in this regard. However, for us, a question remains as to what constitutes the most appropriate set of assumptions for this method. The reviewer states that 'the only way that the approach is justifiable is by assuming ignorability', but would this not suggest that other methods with similar motivations (e.g. CEVAE and TEDVAE) are also 'unjustifiable' based on their stated motivation relating to the inference of confounders from noisy proxies? We would appreciate any further feedback regarding the reviewer's position in light of CEVAE and TEDVAE's seemingly equivalent predicament.
> This notwithstanding, we believe our proposed amendment aligns with the reviewer's suggestion, which is to (1) make the ignorability assumption, (2) discuss the potential benefits of VAEs being able to infer hidden confounders from proxies (particularly in light of the experiments in the CEVAE paper), but make it clear that this requires further assumptions. [see 2nd paragraph page 5].
>
> Finally, we appreciate that AnonReviewer3 may still recommend further clarification, particularly regarding the exposition of the method details, and in addition to our suggested changes  (above) we gratefully welcome additional feedback for how we can improve the clarity of the updated manuscript.

---

### Author Response · Authors · 2020-11-17
**Review References**

For convenience, we have compiled the rebuttal references here:

Alemi et al. (2018) Fixing a broken ELBO. arXiv:1711.00464v3

Louizos et al. (2017) Causal Effect Inference with Deep Latent-Variable Models. 31st Conference on Neural Information Processing Systems

Shalit et al. (2017) Estimating individual treatment effect: generalization bounds and algorithms. Proceedings of the 34 th International Conference on Machine Learning.

Shi et al. (2019) Adapting Neural Networks for the Estimation of
Treatment Effects. arXiv:1906.02120v2

van der Laan, M. J. and Rose, S. Targeted Learning: Causal inference for observational and experimental data (2011) Springer, New York.

Zhang et al. (2020) Treatment Effect Estimation with Disentangled
Latent Factors. arXiv:2001.10652v2

---

### Decision · Program_Chairs · 2021-01-07
**Final Decision**

**Decision:**

Reject

**Comment:**

The paper introduces some good ideas, but I don't think it is quite there in terms of a method to be recommended for publications. I think it is mostly reasonably written (I do not agree with the comment of a 'complete rewrite') but there are indeed some passages for improvement (for instance, an equation as y = σ−1[Q0(t, x)] + εH(t, x)], Section 2, needs comments, as the left hand side is discrete and the right hand side is continuous, unbounded).

My main concern is the disregard for identification. Some citations are unclear (the second-to-last paragraph in Section 4 cites a few papers in identification that have little to do with the problem here, which is proxy-based. The papers cited don't even mention latent variables at all). As stated, the split in three sets of variables as suggested by Figure 1 is just an idealization: there is no reason at all they can be identified, and actually the theory where just Zc is considered impose a lot of restrictions on when we can possible identify Zc (see e.g., Miao et al. 2018, Biometrika, https://arxiv.org/pdf/1609.08816.pdf ). I know that some papers like Louizos et al. play fast and loose with identification too, but at least their Z_c structure they aim at has been studied elsewhere (like the Miao et al. paper), while here, like the Zhang et al. paper cited, may be leading researchers to an unfruitful path. This, combined with the relative modesty of the novelty, is the primary reason for my recommendation. I do think the paper can be improved in a productive way by investigating it from the point of view of either i) the theoretical justification for identification; ii) or from a more empirical direction with much experimentation on the different ways the structured latent space is capturing confounding (the target learning aspect of it is pretty much orthogonal to this).